# Ion Channel Impairment and Myofilament Ca^2+^ Sensitization: Two Parallel Mechanisms Underlying Arrhythmogenesis in Hypertrophic Cardiomyopathy

**DOI:** 10.3390/cells10102789

**Published:** 2021-10-18

**Authors:** Lorenzo Santini, Raffaele Coppini, Elisabetta Cerbai

**Affiliations:** Division of Pharmacology, Department of Neuroscience, Psychology, Drug Sciences and Child Health (NeuroFarBa), University of Florence, 50139 Firenze, Italy; lorenzo.santini@unifi.it (L.S.); elisabetta.cerbai@unifi.it (E.C.)

**Keywords:** HCM, ion currents, afterdepolarizations, excitation-contraction coupling, sarcomeres, EAD, DAD, action potential, potassium, sodium

## Abstract

Life-threatening ventricular arrhythmias are the main clinical burden in patients with hypertrophic cardiomyopathy (HCM), and frequently occur in young patients with mild structural disease. While massive hypertrophy, fibrosis and microvascular ischemia are the main mechanisms underlying sustained reentry-based ventricular arrhythmias in advanced HCM, cardiomyocyte-based functional arrhythmogenic mechanisms are likely prevalent at earlier stages of the disease. In this review, we will describe studies conducted in human surgical samples from HCM patients, transgenic animal models and human cultured cell lines derived from induced pluripotent stem cells. Current pieces of evidence concur to attribute the increased risk of ventricular arrhythmias in early HCM to different cellular mechanisms. The increase of late sodium current and L-type calcium current is an early observation in HCM, which follows post-translation channel modifications and increases the occurrence of early and delayed afterdepolarizations. Increased myofilament Ca^2+^ sensitivity, commonly observed in HCM, may promote afterdepolarizations and reentry arrhythmias with direct mechanisms. Decrease of K^+^-currents due to transcriptional regulation occurs in the advanced disease and contributes to reducing the repolarization-reserve and increasing the early afterdepolarizations (EADs). The presented evidence supports the idea that patients with early-stage HCM should be considered and managed as subjects with an acquired channelopathy rather than with a structural cardiac disease.

## 1. Hypertrophic Cardiomyopathy: A Brief Overview Identifying Sudden Cardiac Death as the Principal Clinical Burden of the Disease

Hypertrophic cardiomyopathy (HCM) is a monogenic inherited cardiac disease that can be caused by more than 1400 different mutations in genes coding for proteins of the cardiac sarcomere or of the adjacent Z-disc [1,2]. The vast majority of pathogenic mutations (largely missense) identified in patients occur in eight sarcomeric genes that were initially associated with HCM by pedigree linkage studies in the 1990s, with just two genes (*MYBPC3* and *MYH7*) accounting for approximately 80% of disease-causing variation [3]. Five percent or less of cases are caused by mutations in Troponin T (*TNNT2*), cardiac Troponin I (*TNNI3*), α tropomyosin (*TPM1*), essential and regulatory myosin light chains and cardiac actin [4,5,6,7]. HCM mutations in the regulatory light chain (RLC) of myosin are quite rare, but their importance is due to their essential structural and functional role, by supporting the architecture of the myosin neck region and finely regulating the kinetics of the actin–myosin interaction [8]. Another HCM-related gene that is becoming increasingly important is TNNC1, encoding Troponin C; notwithstanding that mutations in this gene represent a relatively rare cause of HCM-cases, recent evidence highlights a correlation between TNNC1 variants and an adverse clinical outcome, in terms of earliest onset of the disease and highest frequency of fatal events [9]. HCM shows a prevalence of 1/500 in the general population [1,2], with a worldwide distribution, as cases have been observed in over 60 countries on all continents [2], and it affects males and females [10] as well as subjects of various ethnic origins, with similar prevalence, clinical course and phenotypic disease expression [2,5,10]. The clinical diagnosis of HCM is based on the evidence of a non-dilated left ventricle with an increased wall thickness, without evidence of any other cardiac or systemic disease that might explain the observed hypertrophy, detected with echocardiography [10,11,12] or cardiovascular MRI [13,14]. Histologically, hypertrophic cardiomyopathy is characterized by multiple alterations, such as myocyte disarray (i.e., a deep alteration of myocyte alignment that lose the physiological parallel orientation) [1], interstitial fibrosis [15] and microvascular dysfunction, the latter being the most important substrate for ischemia in HCM hearts [16,17]. These abnormalities concur to promote the creation of specific regions of impaired conductance (reentry circuits), directly contributing to the genesis of the enhanced arrhythmogenic activity (i.e., ventricular tachycardia) in HCM hearts [18], representing the main clinical burden in diseased patients. In fact, HCM is associated with heart failure and atrial fibrillation, yet the most unpredictable and devastating consequence of this disease is sudden cardiac death (SCD) due to primary ventricular tachycardia/fibrillation (VT/VF) [1,2], occurring in 0.5 to 1% of HCM patients per year [19], and particularly affecting children and young adults (age <30 years) [20,21,22]. In HCM, SCD shows an evident relationship with patient age, showing the highest incidence during adolescence and young adulthood [21]. Although SCD risk tends to attenuate with aging, becoming uncommon over the age of 60 years, absolute immunity to SD is never achieved in HCM [23]. Although SCD represents a relatively rare phenomenon in HCM cohorts (<1%/year), the number of known mutation carriers is increasing in the recent years, due to the wider availability of clinical/genetic screening programs in the families of HCM patients [24]. This highlights the need to identify and validate novel risk factors to estimate the arrhythmic propensity of HCM patients, based on their mutation, clinical features and instrumental findings. The early identification of patients at risk remains a very challenging task for the therapeutic management of HCM, as currently available risk-assessment algorithms can identify high-risk patients at an advanced disease stage but are incapable of pinpointing high-risk subjects of young age and with mild disease expression. Indeed, the risk-assessment algorithm contained in the current HCM guidelines [25] highlight clinical and instrumental markers of advanced disease, such as massive hypertrophy or extended replacement fibrosis, as the main clinical correlate of arrhythmic risk. Notably, current risk assessment algorithms have a high predictive value when estimating the presence of advanced structural pro-arrhythmic substrates comprising severe myocardial disarray, fibrosis and ischemia, which are linked with the generation of multiple self-maintaining reentry circuits. However, a vast number of SCD cases occur in young asymptomatic patients with relatively mild disease expression, prompting clinical and preclinical researchers to identify novel clinical markers of arrhythmic risk, in order to select patients who will benefit from undergoing targeted arrhythmia-prevention therapies, either with drugs or with implantable devices. To satisfy this need, many studies have been drawn up through the years to gain deeper insight into the cellular and molecular mechanisms underpinning the pathogenesis of life-threatening arrhythmias in HCM patients with mild hypertrophic phenotype and no major fibrotic scars, using various tools and models to study the underlying electrical mechanisms. Different tools have been used in the recent years to model HCM [26,27,28,29,30,31,32,33,34] in order to unveil the different pathomechanisms contributing to the arrhythmogenic substrate in HCM. Notwithstanding that the aforementioned histological abnormalities represent well-established mechanisms contributing to the arrhythmogenic substrate in advanced HCM, at earlier stages of the disease, when the classical structural abnormalities are much less prominent, different arrhythmic mechanisms originating within the affected cardiomyocytes must be the main cause of ventricular tachycardia in these patients. In this review, we will therefore focus our attention on the cellular mechanisms of arrhythmias in HCM hearts. In particular, we will provide a detailed description of the profound electrical impairment at the level of the cardiac cell, analyzing the altered regulation of action potential (AP) and reviewing the principal ion current alterations involved. Moreover, the electrophysiological (EP) impairment of the cardiac cell will be further explored focusing on intracellular Ca^2+^ handling abnormalities, observed consistently in animal models of HCM and human samples [22,26,35,36,37,38]. We will then focus on the consequences of the increased myofilament Ca^2+^ sensitivity, frequently observed in models of HCM-related mutations, where an increased cytosolic Ca^2+^ buffering capacity due to a slower Ca^2+^ dissociation from myofilaments leads to an altered kinetics of calcium transients. Finally, we will elucidate the various pathogenic mechanisms by which the altered regulation of ion currents and the increased myofilament Ca^2+^ sensitivity directly promote the enhanced arrhythmogenic activity in hearts with early-stage HCM.

## 2. Changes in the Density of Ion Currents Result into Action Potential Prolongation in HCM Cardiomyocytes: Insights from Multiple Models

The cardiac ventricular AP comprises 5 sequential phases (0–4), each of them tightly regulated by the voltage- and time-dependent opening and closure of sodium, calcium, and potassium channels. During the diastolic phase of the AP, a prominent role is played by inward rectifying K^+^ (I_K1_) channels, whose open state directly determines the resting membrane potential, while voltage-sensitive Na^+^ and Ca^2+^ channels are closed. The arrival of the depolarizing electrical stimulus from a neighboring cardiomyocyte promotes the immediate opening of the inward Na^+^ current (I_Na_), thus initiating a fast depolarization of the membrane potential (phase 0). The depolarization phase is immediately followed by a rapid and short-lasting repolarization (phase 1), characterized by the concomitant closure of Na^+^ channels and activation of the transient outward potassium current (I_to_). This initial depolarization is abruptly interrupted by the quick inactivation of I_to_, which corresponds to the start of the plateau phase (phase 2). The plateau consists of the counteracting action of the inward Ca^2+^ current mediated by the L-type calcium channels (I_CaL_) and the outward K^+^ current flowing through the rapidly (I_Kr_) and slowly (I_Ks_) activating delayed-rectifier K^+^ channels. These opposing currents result in an “equilibrium phase” characterized by a very small change of membrane potential, which remains stable at around 0mV for 150–250 ms (phase 2). The repolarization is then completed when I_CaL_ is time-dependently inactivated, thus letting repolarizing K^+^ currents (I_Kr_, I_Ks_, and I_K1_) bring back the membrane potential to negative values (phase 3), until the resting membrane potential is reached. The alteration of a single ion current involved in the process could deeply impair the morphology and kinetics of the ventricular AP. Evidence suggests that the electrophysiology of the cardiomyocyte is deeply impaired in HCM, with alterations that can be observed for nearly every current contributing to the AP [26,32], consequently leading to marked abnormalities of AP kinetics.

### 2.1. Human Surgical Samples

Among the different platforms that have been used to investigate the cellular pathways involved in the pathomechanisms of cardiac diseases, human cardiomyocytes isolated from fresh surgical samples are the most informative [39]. For this reason, to better identify the EP impairment of HCM myocardium and gain deeper insight into specific mechanisms, our group recently focused on investigating ventricular cardiomyocytes isolated from surgical samples of HCM patients who underwent surgical septal myectomy due to severe LV outflow tract obstruction [26,40]. Our group has access to surgical myectomies from HCM patients, used to isolate fresh viable cardiac cells, suitable for patch clamp recordings. These cells are compared to cardiac cells obtained from samples of non-failing/non-hypertrophic surgical patients [26,40]. Our analysis highlighted marked changes of the ion currents in HCM ventricular cardiomyocytes (Figure 1). While L-Type Ca^2+^ current (I_CaL_) and late Na^+^ current (I_NaL_) were both significantly increased in HCM cardiomyocytes, the inward-rectifier K^+^ current (Kir2.1/I_K1_), the transient outward K^+^ current (Kv4.3/I_to_) and the delayed rectifier K^+^ currents (I_Ks_, I_Kr_) showed a marked reduction [26,41]. The concomitant perturbation of inward and outward currents resulted in a marked alteration of the AP profile, whose repolarization phase is slower compared to APs recorded in control cardiomyocytes. APD recorded in HCM cardiac myocytes is markedly prolonged compared to ventricular cells obtained from the septum of non-hypertrophic patients with aortic or mitral valve disease [26,41]. The increased I_NaL_ significantly contributes to the prolongation of APD, as confirmed by the marked shortening of the AP duration obtained when exposing HCM cells to I_NaL_ blockers such as ranolazine [26,40]. Among the different molecular mechanisms underlying ion channel alterations in human HCM myocardium, the increased activation of Calcium-Calmodulin Kinase II (CaMKII) appears to play a major role [42,43,44,45,46,47,48,49]. Details on the role of CaMKII as a determinant of EP abnormalities in HCM are found in Section 4.1.

The prolongation of AP kinetics we observed in human samples was also confirmed by Barajas-Martinez et al. in patch-clamped HCM cardiac cells from a patient with HOCM: APD was significantly longer in HCM cardiomyocytes compared to a typical control myocardial cell and the increased I_NaL_ appeared to play a leading role in the impairment of AP kinetics [27]. All in all, we observed that the alterations of AP shape and duration, combined with the abnormalities of Ca^2+^ transient kinetics and diastolic [Ca^2+^], are responsible for an increased likelihood of early and delayed afterdepolarizations (EADs and DADs, respectively) in HCM cardiomyocytes when compared with controls [26,40,41].

### 2.2. Animal Models

The undoubted translational value of human samples from HCM patients undergoing surgical myectomy is associated with several restrictions (high genotypic variability, scarce availability, wide influence of environmental factors and uneven clinical disease expression) [50] strongly limiting their application to the scientific research [42]. Moreover, all HCM patients who undergo myectomy operations have already reached a relatively advanced disease stage, as symptomatic obstruction rarely develops early in the clinical history of HCM. Therefore, while studying human samples we are completely missing the first stages of the disease, when myocardial structural remodeling is minimal and only cardiomyocyte functional changes have occurred. For these reasons, pathogenic pathways of HCM have been investigated in transgenic animal models that express clinically relevant HCM-associated mutations. Rodent models are most commonly used in the field of disease modeling, including HCM research [39]. Considering that most HCM-causing mutations occur in the *MYBPC3* gene [3,51] and all lead to reduced expression of the MyBPC protein, MyBPC knockout (KO) mice have been frequently used to mimic severe HCM phenotypes [52,53,54,55]. In cardiomyocytes from MyBPC-KO mice, Zhang and coworkers observed an electrical impairment similar to that we observed in human ventricular cardiomyocytes from HCM patients. In particular, the decreased density of repolarizing K^+^ current (normalized to cell capacitance) recorded in KO myocytes resulted in prolongation of AP kinetics at all frequencies of stimulation compared to control (CTRL) cardiac cells. As in human surgical samples [26], the reduction of K^+^ current was also confirmed by quantitative RT-PCR analysis, which highlighted the decrease of the mRNA levels of several K^+^ channel transcripts in KO mouse hearts compared with WT mice. In particular, this reduction was particularly evident for I_to_ [28]. The EP features of a Mybpc3-mutant knock-in HCM mouse model (c.772G>A) were studied by Flenner and coworkers. Measurements with sharp microelectrodes in the LV endocardium revealed that hearts from homozygous mutant mice were characterized by significantly longer APs compared to heterozygous mutant mice and WT mice, both in postnatal day 1 and 30-week-old mice, confirming that: 1) AP prolongation represents one of the main EP abnormalities of HCM and 2) the electrical impairment typical of HCM occurs early during the pathological phenotype development. Interestingly, all outward currents were reduced in aged homozygous mice compared to WT animals and this decrease was particularly evident for I_to_ and I_Ks_ [29]. The reliability of these measurements was confirmed by incorporating all the data related to K^+^ and Ca^2+^ currents in a computational model of a mouse ventricular AP: the “in silico” representation resulted in a similar prolongation of APD in computed homozygous *MYBPC3* mutant cells, as compared to WT myocytes [29].

Together with *MYBPC3*, mutations in β-cardiac MHC (myosin heavy chain) are about 80% of pathology-related mutations in human HCM. Such mutations were studied in rodents by altering the α-cardiac MHC gene; indeed, in rodents, the α-cardiac MHC is the prevalent myosin isoform in the ventricles, while the β isoform is prevalent in larger mammals. In this regard, Hueneke and coworkers [31] explored the electrical remodeling of a αMHC403/+ mouse line (carrying the human MHC mutation Arg403Gln). In patch clamp experiments, the alteration of several K^+^ currents was evident in different regions of the heart, with a reduction in the amplitude of I_to_ and I_Ks_ in the interventricular septum, LV apex, and LV free wall myocytes isolated from 10–12-week-old male αMHC403/+, compared with age-matched WT mice. Moreover, these alterations were not uniform: the density of I_Ks_ was more pronounced in myocytes from the interventricular septum as compared with the LV free wall or the apex, while I_to_ density was lower in the septum, suggesting that a single point mutation is likely to disrupt the physiological regional distribution of different type of ion currents across the LV [31]. One of the main advantages of rodent models is the possibility to study pathological changes during the development of the disease at different stages of disease progression. Our group recently investigated the electrical and EC-coupling changes occurring in a transgenic HCM model carrying the R92Q mutation in the troponin-T gene [56]. We observed that the development of the complete cardiac cellular disease phenotype occurred at approximately 6 months of age and comprised increased late Na^+^ current density, lower expression of K^+^ channel genes, slower Ca^2+^ transients, elevated diastolic Ca^2+^, higher rate of diastolic Ca^2+^ sparks [41] and an increased occurrence of spontaneous diastolic depolarizations. In line with our results obtained in human samples, such changes were associated with an increased activation of CaMKII in ventricular myocardium. In young mice of 4 weeks of age, instead, while most of the changes observed in older mice were still absent, we already observed a clear increase of the I_NaL_. We concluded that the increase of I_NaL_ occurs relatively early during the HCM-related cardiomyocyte remodeling process, and may represent a driver of further pathological changes, amplified by the enhanced CaMKII activity. In line with this interpretation, lifelong treatment of mutant R92Q-TnT mice with ranolazine prevented the development of most of the aforementioned functional abnormalities in adult mice, thanks to the early inhibition of the enhanced I_NaL_, which in turn prevented the activation of CaMKII in the ventricles of adult mutant mice [56]. We also observed that the degree of EP cardiomyocyte remodeling is directly dependent on the nature of the disease-causing mutation. We compared the functional features of mice with the R92Q-TnT mutation with those of mice carrying the E163R variant of the same gene [50]. Interestingly, despite a similar electro-mechanical phenotype featuring diastolic dysfunction and enhanced cellular arrhythmogenesis, myocytes from E163R-TnT hearts did not display alterations of Ca^2+^ transient kinetics or increased late Na^+^ current. In E163R mutant cells, the enhanced arrhythmogenicity may be a direct consequence of the altered myofilament function (see below) or may follow secondary alterations of the ryanodine receptor, leading to the observed increase of spontaneous diastolic Ca^2+^ release from the SR [50]. R92Q mutation was studied also in adult left ventricular cardiomyocytes isolated from guinea pig hearts. R92Q cells recapitulated the different pathophysiological features of HCM, including the marked Ca^2+^ handling impairment and the increased myofilament Ca^2+^ sensitivity of force development. Using this HCM model, Sparrow and coworkers demonstrated for the first time the ability of Mavacamten (MYK-461, a novel myosin inhibitor) to revert the impairment of Ca^2+^ handling and the myofilament Ca^2+^ sensitization mediated by thin filament HCM-causing mutations. Moreover, using Ca^2+^ fluorescent probes able to distinguish [Ca^2+^] in distinct cytoplasmic and myofilament-specific pools, MYK-461 was noticed to reduce cytoplasmic peak systolic fluorescence in the cytoplasm and at the myofilament in R92Q cardiomyocytes. The latter results suggest that MYK-461 may ameliorate diastolic function by directly reversing the specific molecular and cellular modifications provoked by the HCM mutation, as the marked Ca^2+^ handling alteration [57]. Regardless of the mutation, all rodent models of HCM have the same two principal limitations. First, cardiac structural phenotype expression is different from the human disease, in that the distribution of hypertrophy is always concentric in mouse models, while it is always asymmetric in humans; moreover, mice carrying HCM mutations tend to shift away from a purely hypertrophic disease at later stages, progressing towards systolic dysfunction and LV dilatation (in the presence of *MYH6* mutations) or towards a severely restrictive disease (*TNNT2* mutant lines) [56]. The second severe limitation is that mice carrying HCM mutations do not show spontaneous ventricular arrhythmias under physiological conditions [56] and are therefore unfit for studying the development of arrhythmias in intact hearts. For these reasons, transgenic minipig models carrying HCM-related *MYH7* mutations have been developed and are currently being studied [58,59,60]. Transgenic minipig might be the optimal model for studying arrhythmogenesis in vivo in HCM.

### 2.3. hiPSC

Rodent models also have several limitations, the most severe being the large differences in cardiac physiology and electrophysiology when compared with humans and other larger mammals [61,62,63]. The development of patient-derived induced pluripotent stem cells (hiPSCs) started a revolution in both basic and clinical research, and hiPSCs currently play a leading role in the study of the pathomechanisms underpinning inherited genetic heart diseases, such as HCM [64]. HiPSCs can provide an endless source of material [35] and can be potentially differentiated into any cell type: this is particularly important for cells such as cardiomyocytes, which have a limited regenerative potential and cannot be easily kept in culture after isolation from adult heart samples. Therefore, hiPSC-derived cardiomyocytes (hiPSC-CMs) have been extensively used to gain deeper insight into different pathomechanisms of HCM, including the electrical abnormalities [32,33,34]. The EP behavior of HCM iPSC-CMs carrying a single missense mutation (R442G) in the *MYH7* gene was initially investigated by Han and coworkers, showing that this line precisely reproduced the different features of the human disease, such as impaired sarcomere structure, increased cell size and most of the electrical alterations observed in adult human HCM cells. In particular, they observed a clear prolongation of the APD in HCM cells (evident at both 90% and 50% repolarization, APD90 and APD50) associated with an increase of Ca^2+^ currents, as compared with control hiPSC-derived CMs. In addition, patch-clamp experiments revealed electrical abnormalities in Na^+^ and K^+^ currents [32]. The prolongation of APD and the related alteration of I_CaL_ density was evidenced also by Prondzynski and coworkers, who casted mutant hiPSC-CM (c.740C>T; p.T247M in the *ACTN2* gene) into engineered heart tissues (EHTs), combining a classical cell culture method with an innovative tridimensional approach. The increased I_Ca,L_ in mutant compared to control cardiomyocytes resulted in a significant modification of the repolarization phase of AP kinetics: sharp microelectrode recordings performed in intact EHTs highlighted a prolongation of APD50 and APD90 in association with an increased AP amplitude [33]. Jehuda and coworkers [34] studied a hiPSC-CMs line carrying the R302Q mutation in the PRKAG2 gene (encoding the γ-subunit of adenosine monophosphate kinase, AMPK). In particular, the pathological line was compared to an isogenic control substrate obtained through the correction of the specific genetic defect in the pathological line by the application of the CRISPR-Cas9 technique. Patch-clamp experiments on single hiPSC-CMs showed that the pathological cells were characterized by prolonged APD when compared with the isogenic control line [34]. This result revealed that the EP impairment of the pathological line is totally reverted by the CRISPR mediated correction of the PRKAG2 mutation, confirming the profound direct correlation between HCM-causing mutations and the EP abnormalities of HCM myocardium. The correlation between R92Q mutation and the EP impairment was also validated in hiPSC-CMs carrying a heterozygous R92Q mutation R92Q-hiPSC-CMs, which recapitulated the different mechanical and EP properties of HCM, displaying increased force generation and an impairment of AP kinetics and Ca^2+^ current. The R92Q mutation directly influenced the positioning of tropomyosin along the thin filament, reducing the proportion of tropomyosin lying in the blocked (inhibitory) state and increasing the population of tropomyosin in the closed state, leading to an activation of the thin filament at low calcium levels. Indeed, the molecular mechanisms below the R92Q mutation is a decreased population of the thin filament blocked state, promoting higher force generation during calcium-based activation. The latter represents the initial molecular insult, which leads to multiple downstream alterations, from the expression of genes associated with Ca^2+^ handling to EP impairment [65].

## 3. Myofilament Ca^2+^ Sensitization as a Determinant of Diastolic Dysfunction and Arrhythmogenesis in HCM Hearts

### 3.1. Increased Myofilament Ca^2+^ Sensitivity in HCM

Calcium-induced calcium release (CICR) is a process composed of subsequent and coordinated steps that starts with the initial membrane depolarization leading to Ca^2+^ entrance into the cell through L-type Calcium channels [66]. This increase of intracellular Ca^2+^ concentration promotes a massive release of Ca^2+^ from the sarcoplasmic reticulum by the opening of ryanodine receptors (RyRs) (shift of intracellular Ca^2+^ from 0.15 µM during diastole to a peak of 1.6 µM [Ca^2+^]). Calcium ions released in the cytosol will then interact with sarcomeric proteins, thus mediating the activation of contraction and the consequent shortening. Thin-filament proteins are profoundly involved in the excitation-contraction coupling. Troponin is a regulatory protein composed of three domains (Troponin I (cTnI), Troponin C (cTnC) and Troponin T (cTnT)) that allow the interaction between myosin and actin and the consequent creation of cross-bridges and force through removal of the tropomyosin obstacle on actin filament upon binding of calcium ions. In particular, cTnI acts as a regulatory unit, mediating the inhibition of the actin filament by tropomyosin; cTnC directly interacts with Ca^2+^ and acts as a Ca-dependent switch of the thin filament; finally, cTnT coordinates the position of Tropomyosin (Tm) on the actin filament [67] and indirectly affects the affinity of cTnC for calcium ions. The increase of cytosolic [Ca^2+^] promotes a rapid increase of the amount of Ca^2+^ bound to cTnC, which induces a conformational modification of the troponin complex that removes the tropomyosin block from the actin filaments, allowing the cross-bridge formation and the consequent force generation [68]. The relaxation process takes place when cytosolic [Ca^2+^] decreases, causing cTnT-bound Ca^2+^ to detach from the troponin complex, which shifts back its resting conformation, moving tropomyosin back to its actin-blocking position, thus stopping acto-myosin interactions. The fine regulation of the different steps of this process suggest that mutations in genes encoding for one of the proteins involved could perturb the physiology of the cardiac contraction and relaxation, as well as of the regulation of intracellular Ca^2+^ cycling. Notwithstanding the high genotypical and phenotypical variability characterizing HCM, impaired relaxation has been identified as a common feature in human [69,70,71,72,73] and HCM animal models [37,38,74,75,76,77,78,79,80]. Multiple studies investigating the molecular mechanisms of HCM reveal that increased myofilament Ca^2+^-sensitivity represents a common dysfunction in HCM models and has been proposed as a possible initial trigger of disease pathogenesis [38,79,81,82,83,84,85,86,87,88]. In particular, increased myofilament Ca^2+^-sensitivity seems to underpin the altered myocardial relaxation observed in HCM, directly causing a delay of the onset of ventricular relaxation, thus impairing the physiological LV filling [81,82]. The tendency of sarcomeric mutations to sensitize myofilament to Ca^2+^ was observed in several rodent models of HCM. The correlation between HCM-causing mutations and increased myofilament sensitivity of force development was also demonstrated by computation models: multiscale explicit 3D models precisely described the mechanisms below the ability of L48Q mutation to affect contractile properties of the sarcomere, increasing myofilament Ca^2+^ sensitization [89]. Prabhakar and coworkers demonstrated that an α-Tropomyosin mutation (substitution Glu180Gly at codon 180) altered both diastolic performance and the relaxation phase through a consistent increase of myofilament Ca^2+^ sensitivity [85]. The group of L. Carrier evaluated the consequences of a MYBPC3 mutation (insertion of a G > A transition on the last nucleotide of exon 6) on diastolic function in a mouse model of HCM. Skinned cardiac trabeculae isolated from 7–8-week-old transgenic mice showed an impaired mechanical function, displaying a higher myofilament Ca^2+^ sensitivity. Interestingly, these myofilament abnormalities preceded the development of left ventricular hypertrophy (LVH) in the transgenic rodent model, suggesting that the sensitization of myofilament to Ca^2+^ occurs early during the development of the disease and represents a direct consequence of sarcomeric-causing mutations [38]. The link between myofilament Ca^2+^ sensitivity and HCM development was deeply analyzed also in cardiac surgical biopsies from HCM patients carrying different mutations on the MYBPC3 gene, revealing an increase of myofilament Ca^2+^ sensitivity compared to specimens from nonfailing donors. Moreover, the maximal force development was significantly reduced in MYBPC3 mutants compared with donors [87,88]. Sequeira and coworkers studied several other HCM-causative mutations in different sarcomeric genes, including thick (MYH7, MYBPC3) and thin (TNNT2, TNNI3, TPM1) filament proteins, compared with sarcomere mutation-negative HCM patients and nonfailing donors. Skinned cardiomyocyte force recordings revealed that HCM-mutant samples produced a significantly lower maximal force compared to donor specimens and the sensitivity of myofilaments to Ca^2+^ was increased [86]. While increased myofilament Ca^2+^ sensitivity appears to be a common mechanism occurring in different HCM-related mutations, specific mutated sites promote a more consistent increase of myofilament Ca^2+^-sensitivity. Schober and coworkers focused on different mutations located on the TNNT2 gene (TnT-I79N, TnT-F110I, TnT-R278C mutants): while TnT-I79N mediated a consistent increase of myofilament Ca^2+^ sensitization [90] in association with an increased diastolic [Ca^2+^], TnT-F110I and TnT-R278C were characterized by intermediate and no Ca^2+^ sensitization [82,90], respectively [79]. In line with that, we observed that myofilament Ca^2+^ sensitivity was greatly enhanced in the ventricular myocardium from the R92Q-TnT mutant mouse line, while it was unchanged in the E163R-TnT line [50]. Interestingly, mice carrying thin filament mutations associated with a greater increase of myofilament Ca-sensitivity were also characterized by an enhanced tendency towards cellular arrhythmias and diastolic electrical instability [50,90]. It is to be noted that the development of myofilament Ca^2+^ sensitization during HCM pathogenesis may not be univocal, since it can be directly mediated by the mutations in sarcomeric proteins but can also be promoted by secondary disease-related mechanisms. In fact, increased sensitization to Ca^2+^ evaluated in surgical myectomy samples of HCM patients harboring MYBPC3 mutations correlated with low phosphorylation levels of cTnI and cMyBP-C, proteins that are actively involved in modulating Ca^2+^ binding to the troponin complex; indeed, both proteins are specific targets of the β-adrenergic signaling pathway via protein kinase A (PKA) [87,88]. Since the increased myofilament Ca^2+^ sensitivity observed in these cardiac surgical samples from HCM patients was consistently reduced and brought back to the level of non-failing donors by treatment with exogenous protein kinase A (PKA), the increased sensitization to Ca^2+^ is likely not to be directly caused by the specific disease-related mutation but seems to be mediated by secondary disease-related post-translational modifications, as an insufficient PKA-mediated phosphorylation of sarcomeric proteins [87,88]. The involvement of PKA signaling pathway in the impairment of myofilament Ca^2+^ sensitization was investigated also in a rodent model of HCM, carrying a single homozygous mutation (cTnI-R21C) in cTnI. This mutation, associated with HCM and SCD in humans, causes hypertrophy and impairs diastolic function in mice. Biochemical investigations revealed that cTnI-R21C totally prevents PKA-mediated phosphorylation of Ser-23/24 of cTnI in R21C mice, suggesting potential consequences on Ca^2+^ detachment from cTnC and Ca^2+^ sensitivity of force generation. Indeed, the absence of phosphorylation on cTnI caused excitation-contraction uncoupling in R21C mice: Isoproterenol administration (activating the downstream PKA-related pathways) in R21C-isolated cardiomyocytes caused an increase of diastolic [Ca^2+^] that was not associated with shorter diastolic sarcomere lengths, suggesting that the alteration of a single intracellular pathway is likely to indirectly affect myofilament Ca^2+^ sensitization, thus delaying muscle relaxation [91]. Sequeira and coworkers observed that the sensitization of myofilaments to Ca^2+^ is a reversible but mutation-specific process. In fact, while the increased sensitization to Ca^2+^ observed in myectomy samples from patients carrying mutations MYBPC, TNNI3, TPM1 gene was reversed after PKA treatment, mechanical abnormalities could not be reverted by PKA in cardiac samples harboring MYH7 or TNNT2 mutations [86].

### 3.2. Myofilament Ca^2+^ Sensitization as a Mechanism for Arrhythmias in HCM

Recent evidence suggests that the increased sensitivity of myofilaments to Ca^2+^ represents an independent pathway for the generation of arrhythmias in early-stage HCM (Figure 2). This concept arises from multiple observations in cardiac samples from patients with heart failure [92] and animal models of myocardial infarction [93]: the presence of increased myofilament Ca^2+^ sensitivity correlated with the presence of ventricular tachycardia and SCD. Moreover, HCM patients carrying cardiac troponin T (TNNT2) mutations (such as TnT-I79N) were identified to be particularly susceptible to SCD even in the absence of the extended fibrosis or massive hypertrophy [94]; as TnT mutations are commonly associated with increased myofilament Ca^2+^ sensitivity, such observations confirm the link between myofibril Ca^2+^ sensitization and ventricular arrhythmias [76]. The cellular pathways correlating increased myofilament Ca^2+^ sensitivity with arrhythmogenicity were first investigated in rodent models carrying TnT mutations that lead to different degrees of myofilament Ca^2+^ sensitization by the group of B. Knollman: TnT-I79N and TnT-F110I promoted a consistent Ca^2+^ sensitization, while the TnT-R278C mutation did not affect myofilament Ca^2+^ sensitivity [76]. Interestingly, the TnT mutant mouse lines used in this study did not show any cardiac fibrosis or hypertrophy. ECG recordings in vivo in mutant and CTRL mice revealed that the occurrence of ventricular arrhythmias was not increased in TnT mutants at baseline. However, β-adrenergic stimulation consistently increased the rate of premature ventricular complexes (PVC) in TnT-I79N and TnT-F110I mice. The in vivo results were then confirmed in Langendorff-perfused hearts loaded with the di-4-ANEPPS voltage-sensitive fluorescent dye: fast pacing frequencies frequently induced sustained ventricular tachycardia in the hearts from TnT-I79N and TnT-F110I mice, as compared with CTRL mice and with animals carrying the non-sensitizing mutation (TnT-R278C). The mechanism underlying the induction of ventricular tachycardia in Ca^2+^-sensitized animals was a slowing down of conduction velocity that occurred at fast pacing frequencies, mediating a dispersion of the excitation wave and promoting the establishment of functional reentry mechanisms in TnT-I79N and TnT-F110I mice. The mechanisms leading to the reduction of conduction velocity and reentry arrhythmias in Ca^2+^-sensitized hearts were explored by investigating the effects of myofiber Ca^2+^-sensitization on AP kinetics and morphology [95]. While WT hearts or hearts carrying the non-sensitizing R278C mutant show a preserved AP morphology, APs recorded in TnT-I79N hearts maintain the overall physiological duration but are characterized by a shortened APD measured at 70% of repolarization (APD70). The shortened APD70 leads to a pathological triangular shape of the ventricular AP, with a reduced ERP that promotes reentry mechanisms [37,82]. The shortening of APD70 was found to be a consequence of the effects of myofiber Ca^2+^-sensitization on Ca^2+^ transients. Indeed, transients recorded in I79N-TnT myocytes were characterized by a significantly smaller amplitude compared to CTRL cells, in association with slower kinetics of decay. Given the reduced amplitude of Ca^2+^ transients in I79N-TnT cardiomyocytes, the forward mode of the Na^+^-Ca^2+^ exchanger, NCX (that is, removal of Ca^2+^ from cytosol), may be consistently reduced, with a net reduction of inward current during the last phase of the AP (APD70-APD90), corresponding to the peak of the transient; this mechanism mediates the observed triangulation of the AP. On the other hand, the terminal repolarization of AP (APD90) is maintained due to the slowing down of Ca^2+^ transient decay kinetics occurring as a consequence of the slower Ca^2+^ detachment from the myofilaments in mutant cardiomyocytes [37]. The key aspect of this novel arrhythmogenic mechanism is that the link between increased myofilament Ca^2+^ sensitivity and the establishment of reentry circuits occurs in the absence of consistent anatomical obstacles, suggesting that arrhythmias may occur in HCM hearts even in the absence of extensive morphological changes. The role of increased myofilament Ca^2+^ sensitivity as a novel arrhythmia-promoting mechanism has been confirmed by using the Ca^2+^ sensitizer EMD-57033, which acutely promoted the occurrence of sustained VT during fast stimulation frequencies in the hearts of WT mice, by replicating the AP modifications evidenced in I79N-TnT hearts. On the other hand, Blebbistatin, a non-specific Ca^2+^-desensitizing compound [96], counteracted the arrhythmia-generation mechanism in TnT transgenic hearts.

The effect of TnT-I79N on myofilament Ca^2+^ sensitivity and the consequent establishment of an arrhythmogenic substrate was reproduced also in human cardiomyocytes by the group of Knollman, who used the CRISP-Cas9 technology to generate TnT-I79N hiPSC-CMs from dermal fibroblasts. TnT-I79N hiPSC-CMs replicated in vitro the principal features of HCM, such as the disorganized sarcomere organization, the increased cardiomyocyte contractility and the impaired relaxation. Moreover, since TnT interacts with TnC in the Troponin complex and TnC binds a consistent amount of Ca^2+^ in the cytosol, mutations in TnT are supposed to impair cytosolic Ca^2+^ binding affinity; as expected, TnT-I79N increased the affinity of TnC to Ca^2+^, impairing Ca^2+^ buffering properties of the cardiac cell and consequently mediating a reduction of Ca^2+^ transient amplitude. The impaired cytosolic Ca^2+^ buffering in I79N CMs directly have an impact on AP morphology, which acquires a “triangulated shape”. The effect of the reduced Ca^2+^ amplitude on AP morphology can be explained considering that Ca^2+^ extrusion via the Na-Ca exchanger (NCX) promotes an inward current that contributes to the phase 3 of the AP. A lower Ca^2+^ transient amplitude likely generates reduced NCX currents, thus shortening the early repolarization of AP (APD30, APD50 and APD70), while late repolarization (APD90) is comparable to CTRL CMs. This evidence was confirmed by replacing extracellular Na^+^ with Li^+^: while Li+ enters the cardiac cell through Na^+^ channel, Ca^2+^ extrusion via NCX is arrested, thus removing the discrepancies in APD30, APD50 and APD70 between CTRL and HCM CMs. The leading role played by increased myofilament Ca^2+^ sensitization in AP shape remodeling was confirmed by the use of two specific compounds: While blebbistatin (Ca^2+^ de-sensitizer compound) prevented the AP remodeling of TnT-I79N hiPSC-CMs, EMD57033 (calcium sensitizer) administration replicated the consequences of increased myofilament Ca^2+^ sensitivity effect in control CMs [97,98]. Besides the effect of increased myofilament Ca^2+^ sensitivity on AP shape, the altered Ca^2+^ transient amplitude and kinetics following the higher cytosolic Ca-buffering capacity may further contribute to promoting arrhythmias. TnT-I79N cardiomyocytes showed slower kinetics of Ca^2+^ transient decay and an increased diastolic Ca^2+^ concentration (particularly at high stimulation rates) [79]. The sensitization of TnT-I79N fibers to Ca^2+^ also significantly increased post-pause SR Ca^2+^ content in TnT-I79N myocytes compared to TnT-WT myocytes, leading to Ca^2+^ overload and spontaneous Ca-release from the SR during stimulation pauses [79]. These effects at the cellular level were also replicated in the intact heart, where Ca^2+^ re-uptake from cytosol is slower and end-diastolic cytosolic [Ca^2+^] was found to be increased in TnT-I79N compared to WT and TnT-R278C hearts [79]. In ventricular myocytes from mice carrying the R92Q-TnT mutation, characterized by a marked myofilament Ca^2+^ sensitization, we also observed slower Ca^2+^ transient decay kinetics, associated with a sustained increase of diastolic [Ca^2+^] and an increased probability of spontaneous calcium waves and premature contractions during pauses [50,56]. The increased diastolic cytosolic [Ca^2+^] and the overload of Ca^2+^ within the SR at high pacing rates and during pauses represent an ideal substrate for EADs, DADs and triggered activity. Indeed, the increased myofiber Ca^2+^ sensitivity in TnT-I79N hearts promoted an increased incidence of EADs followed by triggered activity at high frequencies of stimulation, compared to TnT-R278C and WT. Late-phase EADs and DADs are a direct consequence of the increased RyR open probability due to cytosolic and reticular Ca^2+^ overload and might be potentiated by post-translational modifications of the RyR (e.g., increased RyR phosphorylation by the CaMKII), which were observed in human and mouse HCM hearts [26,50,56]. This might be a universal mechanism by which myofiber Ca^2+^ sensitization generates pause-dependent EADs and triggered arrhythmias in HCM hearts. With regards to the impairment of myofilament Ca^2+^ binding affinity and its correlation with an arrhythmogenic substrate, recent findings suggest that a special focus should be given to TNNC1. Besides TNNC1 variants causing an extremely limited number of total HCM cases, the identification of a specific TNNC1 variant (alanine substituted by serine at position 31, A31S) in a pediatric-onset HCM proband opened novel perspectives in the understanding of HCM pathogenesis, revealing the specific mechanisms underpinning the increased sensitivity of myofibers to Ca^2+^ and its direct relationship with the increased occurrence of ventricular fibrillation. In particular, the novel cTnC-A31S variant was identified in a 5-year-old HCM patient with multiple episodes of ventricular fibrillation, where aborted sudden cardiac death occurred before the age of 4. Functional studies performed in cardiac porcine fibers reconstituted with WT cTnC and cTnC-A31S revealed that the cTnC-A31S variant, located in the non-functional Ca^2+^ binding site I, is likely to promote changes in the Ca^2+^ binding capacity of cTnC, causing a marked increase of Ca^2+^ sensitivity that will ultimately result in an arrhythmogenic substrate [99]. This evidence further confirms the need to constantly update and improve current guidelines for the stratification of arrhythmic risk in patients, and in particular the need to identify novel arrhythmic risk factor for early-stage young patients with mild structural disease expression.

## 4. Ion Channel Remodeling of the Cardiac Myocyte as a Major Trigger of Arrhythmogenesis in HCM Hearts

Data from studies in human cardiomyocytes and transgenic mouse models of HCM showed that pathological abnormalities in ion currents and Ca^2+^ handling [26,40,50,56] impair the EP properties of the myocardium and promote the generation of arrhythmic-trigger events at cellular level, such as EADs and DADs. These cellular arrhythmic events can trigger premature spontaneous action potentials in the intact heart, which may be conducted to the surrounding ventricular tissue giving rise to premature ventricular contractions; conducted premature ventricular activations, under certain conditions, represent the initial trigger for the formation of reentry circuits and for the initiation of non-sustained or sustained ventricular tachyarrhythmias. Interestingly, in a large study we conducted in over 25 samples from HCM patients, we observed that cells isolated from samples of patients with a clinical history of documented non-sustained VT had on average an increased likelihood of EADs and DADs [26]. This observation suggests that a larger degree of cellular EP remodeling might be related with an increased risk of arrhythmias in certain patients. The real challenge is to identify simple clinical instrumental correlates of the increased cellular arrhythmogenicity in such patients, without the use of invasive procedures such as catheter EP studies. We are currently performing a large follow-up study on over 70 patients who underwent myectomy, aimed at identifying potential clinical correlates of the enhanced cellular arrhythmogenicity (i.e., the presence of frequent EADs and/or DADs) observed in isolated cells, trying to identify new clinical markers of the risk of VT and lethal arrhythmias.

In this section we will focus on the ion currents contributing to AP, evidencing how their alterations contribute to the development of a cellular arrhythmogenic substrate in the HCM heart. As detailed above (Section 2.1), we performed extensive EP characterizations on cardiomyocytes isolated from surgical samples from over 60 HCM patients, revealing a marked prolongation of APD due to multiple ion current alterations [26,41]. Studies performed by other groups in hiPSC-CMs confirmed our observations [32]. EP characterization of a hiPSC line carrying a single missense mutation (R442G) in the MYH7 gene highlighted a markedly prolonged AP, together with alterations in I_Na_, I_Ca_ and I_K_ currents, similar to those we observed in surgical samples. These abnormalities resulted in an increase of the rate of arrhythmias in hiPSC-CM monolayers. The rate of spontaneous events was further worsened by a 5 day treatment with isoproterenol, which significantly increased the incidence of premature beats and irregular beating rates in HCM hiPSC-CMs [32]. Of note, we previously observed a similar effect of isoproterenol in a transgenic mouse model of HCM carrying the R92Q-TnT mutation [56]. In the same hiPSC-CM model, the involvement of the β-adrenergic pathway and of I_CaL_ was further investigated by treating cell lines with the β_1_-adrenergic blocker metoprolol or with the calcium channel blocker verapamil. Both drugs significantly decreased or totally abolished arrhythmogenic events in HCM hiPSC-CMs [32]. Prodzinsky and colleagues, in a different HCM hiPSC-CM line, showed that I_CaL_ was largely increased in the mutant line and the calcium channel blocker diltiazem reduced the electrical dysfunction of HCM hiPSC-CM and shortened their APs. This result highlighted the important role of I_CaL_ in the EP dysfunction of ACTN2-related HCM and allowed the authors to translate this observation to the clinic by developing a personalized therapy approach. The patients, son and sister of the iPSC-originating proband, were put under diltiazem treatment; this change led to reduction of QTc in both patients, suggesting that the impaired EP phenotype was, at least in part, determined by the increased I_CaL_ [33]. Interestingly, in human cardiomyocytes from myectomy samples, we observed that the kinetics of I_CaL_ inactivation, already slower that controls at baseline, were further slowed down under β-adrenergic stimulation, leading to a paradoxical prolongation of APs when cells were treated with isoproterenol [40,41,49,100]. Such behavior was associated with a marked increase of both EADs and DADs during β-stimulation, highlighting the importance of treating HCM patients with β-blockers to prevent stress-induced arrhythmias.

In addition to the altered I_CaL_ amplitude and kinetics, the increased I_NaL_ plays a leading role in causing the increase of EADs and DADs in HCM ventricular cardiomyocytes, both at baseline and in response to β-adrenergic stimulation. We observed that I_NaL_ also increases in human HCM cardiomyocytes exposed to isoproterenol and pharmacological I_NaL_ inhibition with ranolazine prevents the prolongation of APs and the increase in cellular arrhythmogenesis following β-stimulation [40,49].

Besides its direct effects on AP duration, I_NaL_ is also indirectly involved in the Ca-dependent pathomechanisms of HCM arrhythmogenesis, in that it contributes to the overload of diastolic [Ca^2+^]_i_ in HCM cardiomyocytes. Intracellular Na^+^ directly regulates the function of NCX, the main component responsible for the maintenance of intracellular Ca^2+^ balance. When [Na^+^]_i_ exceeds 12–15 mmol/L (that is, three times normal intracellular sodium concentrations), the Na^+^ gradient becomes insufficient to maintain the regular Ca^2+^ extrusion through the NCX. Increased [Na]_I_ not only causes a reduced forward NCX function during diastole, leading to insufficient Ca^2+^ extrusion, but also potentiates the reverse mode activity of NCX during systole, thus bringing more Ca^2+^ into the cell. The increased diastolic [Ca^2+^]_i_ increases the probability of spontaneous Ca^2+^releases through the ryanodine receptors (RyR) during diastole, facilitating the occurrence of calcium waves and arrhythmogenic DADs. In line with that, inhibition of I_NaL_ with ranolazine or disopyramide not only decreased the rate of EADs but also greatly reduced the occurrence of DADs in human and mouse HCM myocytes [26,40,41], thanks to the normalization of the kinetics of Ca^2+^ transients and the reduction of diastolic [Ca].

In parallel with “wet lab” experiments, computational approaches contributed to unveiling the mechanisms underlying cardiac arrhythmogenesis in HCM, by modeling the electrical properties of the cardiac myocyte. Focusing on the patch clamp and Ca^2+^ imaging data from Coppini and coworkers [26], Passini and colleagues used the original model by O’Hara, Rudy et al. [101] to develop populations of human non-diseased and HCM AP models that consider inter-subject variability of functional parameters. The human HCM population was constructed applying the ion channel changes previously identified in HCM cardiomyocytes, that is, increased I_NaL_ (+165%) and I_CaL_ (+40%) and decrease of repolarizing currents such as I_to_ (−70%), I_K_ (−40%) and I_K1_ (−30%). Moreover, the ion current abnormalities were combined to an increased diastolic [Ca^2+^]_i_ and the slowing down of Ca^2+^ transient kinetics to reproduce the results obtained in wet experiments. Based on these findings, the authors evidenced single and multiple EADs in 752 out of 9118 HCM models, observing that all the models characterized by the presence of EADs showed an altered repolarization reserve due to markedly increased inward currents such as I_CaL_, I_NaL_ and I_NCX_. In particular, computational data highlighted the main role played by overexpressed I_CaL_ in the impairment of the repolarization reserve and the consequent occurrence of pro-arrhythmic events [102].

### 4.1. Role of CaMKII Activation in the Ion Channel Remodeling of HCM Myocardium

One of the main mechanisms underlying the altered electrical properties of the HCM cardiac myocyte appeared to be the increased activity of the Ca^2+^/calmodulin-dependent protein-kinase II (CaMKII), which was observed in animal models and human samples with cardiac hypertrophy and diastolic dysfunction [42,43,44]. Results collected by our group from HCM samples revealed that the auto-phosphorylation of CaMKII (a marker of its activation state) was 3.5-fold higher as compared with controls [26]. Notably, the initial activation of CaMKII can be directly promoted by sarcomeric mutations that increase the sensitivity of myofilaments to Ca^2+^ and/or alter cell energetics and energy-dependent processes such as the reuptake of Ca^2+^ into the SR by the SERCA pump. The consequent increase of cytosolic [Ca^2+^]_i_, especially during the diastolic period, hyperactivates CaMKII. The correlation between the increased activity of CaMKII and the electrical impairment of the HCM cardiac cell can be explained by considering the four downstream targets of this enzyme: ryanodine receptors, L-type Ca^2+^ channels, Na^+^ channels and phospholamban. Auto-phosphorylation of CaMKII enhances its activity [45] on the downstream targets: the phosphorylation of the L-type Ca^2+^ channel β-subunit strongly impacts on the kinetics of I_Ca,L_, slowing down current inactivation [26,40,46]. In turn, this effect prolongs systolic Ca^2+^ entry that contributes to the accumulation of [Ca^2+^]_i_ during the diastolic phase. In a similar manner, the phosphorylation of cardiac Na^+^ channel (NaV1.5) by CaMKII alters the inactivation of I_Na_ [26,47,48], resulting in an increased late Na^+^ current (I_NaL_), which was found to be markedly larger in human HCM myocytes when compared with non-hypertrophic cardiomyocytes. In this sense, the enhanced CaMKII activity could be viewed as a cause of AP remodeling, representing the link between the initial cytosolic Ca^2+^ accumulation and the early ion channel changes [49]. On the other hand, the enhanced CaMKII activity is also a consequence of the early cellular EP remodeling, in that the enhanced I_NaL_ hampers NCX function, leading to further accumulation of diastolic Ca^2+^, which in turn determines an additional stimulation of CaMKII activity [49], generating a vicious circle. In line with that, direct acute inhibition of CaMKII with AIP-II lowered diastolic [Ca] and diastolic tension [49]. Additionally, the sustained hyperactivation of CaMKII may also contribute to the reduction of K^+^ currents; our data from surgical HCM samples highlight a marked reduction of the levels of mRNA coding for K^+^ channel subunits in the myocardium, suggesting that the increased activity of CaMKII in the pathological ventricles could regulate the expression of K^+^ channel subunit at the transcriptional level, likely modulating histone-deacetylase (HDAC) activation and function [26,49]. Therefore, sustained activation of CaMKII can also be one the causes of late-stage cellular EP alterations in HCM, mainly driven by changes of ion channel expression levels. Another strategy to demonstrate that CaMKII hyperactivation is strictly connected to Ca^2+^ overload is the inhibition of phospholamban (PLN). Considering the role of Ca^2+^ handling impairment in the pathogenesis of HCM and the correlation between the latter and the altered diastolic function (main determinant of the different symptoms of HCM patients), Chowdury and coworkers evaluated whether the prevention of the diastolic dysfunction could attenuate or arrest the HCM phenotype development in a transgenic mouse model expressing mutant cTnT-R92Q. To reach this goal, by promoting SR Ca^2+^ re-uptake, TnT-R92Q HCM mice with phospholamban knockout (PLNKO) were developed. PLNKO prevented the development of the HCM phenotype (absence of fibrosis, reduced activation of the hypertrophy-related gene expression program) in R92Q transgenic mice, reducing CaMKII phosphorylation but preserving the increased myofilament Ca^2+^ sensitivity of force development. These results suggest that the absence of PLN removed the inhibition of SERCA2, allowing an increased activity of the protease that despite its reduced function due to increased ATP consumption, can still reuptake Ca^2+^ into SR, preventing the increased diastolic [Ca^2+^] and the consequent vicious cycle mainly driven by CaMKII phosphorylation and hyperactivation, finally leading to normal relaxation [103]. The central role of intracellular Ca^2+^ impairment in the pathogenesis of HCM was also evidenced by Li and coworkers, who developed a knockout Muscle LIM protein (MLP, CSRP3) human embryonic stem cell (hESC) line using a CRISPR/Cas9 technique. MLP represents a main regulator of striated muscle function and the absence of this protein was demonstrated to induce the multiple phenotypic features of HCM, such as morphological (increased cell size, multinucleation) and functional (impaired sarcomere structure) alterations. After 30 days post differentiation, this pluripotent stem cell line developed the main pathophysiological changes of HCM, showing mitochondrial impairment and Ca^2+^ handling properties alteration. The centrality of intracellular Ca^2+^ disruption in the HCM pathogenesis was confirmed by verapamil administration, exerting a beneficial role on HCM H9 cell line: verapamil restored Ca^2+^ homeostasis, subsequently preventing the development of HCM manifestations [104]. Ras associated with diabetes (RAD) is a membrane protein regulating the cardiac L-type Ca^2+^ channels (LTCC). Generating a human embryonic stem cell line (hESCs) with RAD deficiency (RRAD^−/−^) through CRISPR/Cas9 technique, Li et al. observed that RAD absence was responsible for the onset of a HCM phenotype development, with the RAD hESCs line displaying typical phenotypical features of HCM, e.g., increased size of cardiomyocytes, multi-nucleation and impairment of Ca^2+^ transient. In particular, the onset of impaired Ca^2+^ homeostasis preceded the hypertrophic phenotype, suggesting that a Ca^2+^ handling impairment plays a central role in the pathogenesis of HCM. This is further highlighted by a chronic verapamil treatment (10 days, 100 nM), showing that the blocking of L-type Ca^2+^ channel restored the HCM phenotype; while Ca^2+^ handling homeostasis was reverted by normalizing SR Ca^2+^ storage, cell size and p-CaMKII significantly decreased [105].

Besides its effect on ion channels, CaMKII is also involved in the modulation of myofilament Ca^2+^ sensitivity through a direct effect on sarcomeric proteins [106,107]. In fact, when the pacing rate increases, CaMKII-mediated phosphorylation of TnI promotes a frequency-dependent myofilament Ca^2+^ desensitization (FDMCD) and the consequent acceleration of relaxation [108]. While the hyperactivated CaMKII contributes to the impairment of HCM myocardium through several mechanisms, CaMKII-mediated phosphorylation of TnI may at least partially counteract the deleterious functional consequences of CaMKII activation, contributing to the preservation of diastolic function. Moreover, considering that the increased myofilament Ca^2+^ sensitization is associated with electrical impairment in HCM muscle, this effect could partly counteract the arrhythmogenic consequences of the increased myofilament Ca^2+^ sensitivity in HCM. Further investigation of the complex role of CaMKII in HCM muscle is warranted.

Besides the role of increased [Ca^2+^] in the promotion of CaMKII hyperactivation, recent evidence highlighted a novel mechanism linking the increased reactive oxygen species (ROS) production with CaMKII activation in cardiac diseases [109]. ROS are predominantly generated at the level of the electron transport chain (ETC) during the process of oxidative phosphorylation leading to ATP synthesis. During the transfer of electrons from carbon substrates (e.g., fatty acids and pyruvate) to nicotinamide adenine dinucleotide (NADH) and flavin adenine dinucleotide (FADH_2_), complex I (NADH and ubiquinone oxidoreductase) and complex III (Q-cycle, cytochrome bc1 complex and coenzyme Q) of the ETC produce anion superoxide (O_2_^−^) in the mitochondrial matrix and intermembrane space, respectively [110]. Due to its unstable and highly reactive state, O_2_^−^ rapidly generates other ROS, such as hydrogen peroxide (H_2_O_2_) and hydroxyl radicals (HO^−^), which can affect multiple cellular pathways by directly inducing post-translational modifications on lipid and proteins and damaging DNA [111]. Notwithstanding that ROS formation is physiologically compensated by the action of specific antioxidant defenses composed of an enzymatic system (including three superoxide dismutases, catalase and glutathione peroxidase [112]) and a nonenzymatic pathway (antioxidant scavengers, such as vitamins E and C [113]), under different pathological conditions the cell’s antioxidant capacity can be surmounted, allowing the generation of an imbalanced cellular redox state [114,115,116,117,118]. Early in the century, clinical and pre-clinical investigations highlighted an improved mitochondrial production of ROS in the impaired myocardium, suggesting a potential leading role played by ROS and ROS-associated signaling pathways in the pathogenesis of cardiac diseases and heart failure [114,115,116,117,118]. The involvement of an impaired redox balance in HCM pathogenesis has been suggested by preclinical evidence, highlighting increased indices of oxidative stress markers in the heart and blood of HCM animal models [119,120,121,122,123], confirmed by the identification of an impaired redox status in HCM patients [124,125,126]. Multiple indications highlight a correlation between ROS and arrhythmogenesis in HCM. H_2_O_2_ administration promotes an enhanced I_NaL_, as the oxidation of Na^+^ channels mediates a delay of channel inactivation, consequently inducing a prolongation of AP duration and an increased likelihood of DADs [127]. Ros can promote enhanced spontaneous activity also by directly acting on SERCA and ryanodine receptors (RyR), via alteration of the redox-sensitive cysteine residues in specific positions of SERCA [128] and RyR [129]. By directly acting on specific sulfhydryl groups, ROS impair SERCA pump function and destabilize ryanodine receptors [128], promoting diastolic Ca^2+^ leak from the SR [130]. The abnormal increase of diastolic [Ca^2+^] stimulates NCX-mediated electrogenic depolarizing activity, thus leading to DADs [131]. Moreover, the arrhythmogenic substrate can be potentially mediated by the ROS-induced regulation on K^+^ channels, through a reduction of I_K_ densities [132].

However, most of the pro-arrhythmic effects of ROS are mediated by the enhanced activation of CaMKII. Erickson and coworkers discovered a novel mechanism by which oxidative modification of the Met281/282 site (in the vicinity of T287), located in the regulatory domain of CaMKII, enhanced the redox-dependent activation of the kinase, thus supporting a constitutive activity of the enzyme [109]. Exposure of CaMKII to Ca^2+^/calmodulin complex is fundamental for oxidation-mediated CaMKII activation, suggesting that the conformational modification induced by Ca^2+^/CaM interaction may expose a specific residue in the regulatory domain of CaMKII that will be oxidized by H_2_O_2_ [109]. These observations suggest that altered redox balance and impaired Ca^2+^ handling contribute to the pathogenesis of HCM with different pathways, ultimately resulting in the activation of CaMKII, that in turn aggravates the EP impairment of HCM cardiomyocytes. Myeloperoxidase (MPO) is an enzyme taking part in the different processes involved in the generation of ROS. MPO is part of haem peroxidases family, expressed in different cellular types as neutrophils and monocytes. Ramachandra and coworkers revealed that MPO is also expressed in iPSC-CMs from patients with mutations in MYH7 and MYBPC3, highlighting its role in the derangement of the relaxation phase. Chronic treatment (1 week) with MPO inhibitor AZD590413 (10 mM) was demonstrated to significantly reduce Ca^2+^ transient decay phase in both IPSC-CMs lines and alleviated the relaxation defect, suggesting that MPO inhibition improved contractility in HCM-CMs, potentially representing a novel HCM pharmacological strategy. The mechanism below MPO effect consists of the alteration of MYBPC3 phosphorylation status: the phosphorylation status of serine residue 282 (Ser-282) on MYBCP3 was significantly reduced in both IPSC-CMs lines compared to control and were increased after AZD590 treatment [133].

### 4.2. Timeline of Ion Current Remodeling in HCM

Changes of INaL and ICaL appear to be, at least in part, independent from the complex changes of the cardiomyocyte expression program that characterizes the advanced hypertrophic remodeling in HCM and other diseases. As confirmed by the results obtained by us and other groups in young transgenic mice and cell lines, the increase of I_NaL_ and I_CaL_ appear rather early during HCM pathogenesis and may be part of the initial response of the affected myocardium to the disease-causing mutation. In the advanced disease (that is, in human samples), changes of channel expression may appear on top of the aforementioned post-translational modifications. As an example, we observed a slight increase of the mRNA and protein expression of CaV1.2 in HCM but did not observe any changes in the expression of NaV1.5 [26,49]. Whether the abnormal overexpression of neuronal I_Na_ subtypes, such as NaV1.8 [134], contribute to the increase of I_NaL_ in advanced HCM remains to be determined.

On the other hand, our data and results from other models clearly identified the reduction of K^+^ currents as a consequence of changes occurring in the global cellular expression program, tightly linked with the non-specific hypertrophic expression profile that is observed in hypertrophied myocardium due to various causes. Indeed, we found that the reduction of I_to_, I_K1_ and I_K_ at patch clamp experiments is paralleled by a marked reduction of the mRNA coding for all K^+^ channels in human HCM myocardium [26,39,41]. The same pattern of decreased K^+^ channel expression was observed in several transgenic mouse models of HCM carrying different mutations [28,29,30,56]. Toib and colleagues, in a HCM transgenic rodent model with MyBPC knockout (KO), demonstrated that a reduction of K^+^ currents resulted in a significant prolongation of the QT interval at the ECG evaluation and premature ventricular contractions (PVCs) evidenced by telemetry recordings [28]. All in all, K^+^ current decrease appears to be a feature of a more advanced stage of disease, when the full spectrum of hypertrophy-related changes has kicked in and the cardiomyocytes, as well as the intercellular milieu, have already undergone profound structural alterations.

One of the most evident contradictions emerging from the description of HCM-related ion-channel remodeling is the uncommon clinical observation of markedly prolonged QTc interval in HCM patients. The prevalence of long QT (QTc > 500 ms) is below 10% of all HCM patients, still above the healthy population yet quite low if we consider the extensive ion channel remodeling we observed in myectomy samples [135]. A large number of HCM patients have perfectly normal QTc at surface ECG despite having the same features of cellular ion channel remodeling. There are a number of explanations for this apparent discrepancy. Patients who undergo myectomy have on average a marked hypertrophy and a relatively advanced disease. Moreover, the region of the heart that is cut and collected during myectomy operations is the most hypertrophied portion of the ventricles. As a number of changes in ion channels (in particular the expression of K currents) go hand in hand with the general hypertrophic response, we can imagine that the degree of ion channel remodeling is uneven across the heart and goes hand in hand with the asymmetric distribution of hypertrophy. Therefore, areas of profoundly dysfunctional myocytes will be present in the LV alongside regions with nearly normal cells, devoid of electrical abnormalities. This would reduce the global changes of QTc despite the presence of locally abnormal myocytes, especially in the massively hypertrophied upper septum. As we could not provide a direct demonstration of this theory in human hearts, we tried to model this uneven distribution of cellular electrical changes in an “in silico” model of HCM LV developed in collaboration with the group of A. Bueno-Orovio [41]. The model predicted that the minimally affected surface QTc could be explained only by grading the severity of EP abnormalities based on the degree of local hypertrophy. If this was confirmed experimentally, the increased dispersion of repolarization would represent an important contributor to the risk of sustained ventricular arrhythmias in such patients. In this regard, drugs that selectively act on the most remodeled myocardium (such as disopyramide) may exert an additional antiarrhythmic effect in HCM by decreasing the dispersion of repolarization [41].

## 5. Conclusions: Different Electrophysiological Changes at Different Disease Stages

In this paper we reviewed two main classes of mechanisms underpinning the arrhythmogenicity of HCM myocardium (Figure 3). The first class of mechanisms is the complex remodeling of the different ion currents participating in the AP. We started from the description of our data obtained from viable cardiomyocytes isolated from HCM surgical samples, highlighting a profound alteration of both depolarizing and repolarizing currents. In particular, while I_CaL_ and I_NaL_ were significantly increased in HCM cardiomyocytes, I_K1_, I_to_ and delayed rectifier I_K_ currents were consistently reduced compared to non-hypertrophic cells, resulting in a marked prolongation of APD. The increase of I_CaL_ and I_NaL_ is mainly due to post-translational modifications of channel proteins, especially by the increased activation of CaMKII in HCM myocardium, and occur at the very initial stages of HCM pathogenesis, as demonstrated in young transgenic mice and in hiPSC-CMs. On the other hand, the decrease of K^+^ currents depends on changes at expression level, and goes hand in hand with the non-specific hypertrophic expression program that kicks in at later stages of the disease. The slowing down of AP kinetics and the consequent prolongation of APD contributes to the increased rate of EADs, while intracellular Ca^2+^ overload is associated with a higher likelihood of DADs; notably, the presence of EADs and DADs in cardiomyocytes correlated with an history of ventricular arrhythmias in patients. The second arrhythmogenic mechanism highlighted here was described last. The increased sensitization of myofilaments to Ca^2+^, a direct or indirect consequence of HCM-causing sarcomeric mutations, has been associated with an increased likelihood of ventricular arrhythmias in the absence of marked histological abnormalities. This mechanism has been described in detail for mutations occurring in genes encoding thin filament proteins, which are directly involved in the interaction of myofilaments with Ca^2+^, such as TnT. Notably, HCM due to thin-filament mutations has been associated with a worse prognosis as compared with the most common thick-filament mutation [136]. In brief, Ca-sensitized myofilaments promote an arrhythmogenic substrate through two main mechanisms. First, at low frequencies of stimulation (steady state) the increased myofilament Ca^2+^ sensitivity reduces Ca^2+^ transient amplitude and slows down their kinetics, thus inducing a morphological modification of the AP, consisting of a triangular shape that promotes the creation of re-entry circuits through shortening of the local ERP. Additionally, at higher stimulation frequencies, the increased cytosolic Ca buffering causes a slowing of Ca^2+^ transient decay and diastolic Ca^2+^ accumulation, facilitating spontaneous Ca^2+^ release from the SR, which in turn promotes late-phase EADs and DADs. Current arrhythmia-preventing strategies in HCM are targeted only towards patients with advanced disease, where massive hypertrophy, microvascular ischemia and large fibrotic scars are the main sources responsible for arrhythmias. Such patients are managed exactly like patients with advanced heart failure due to secondary causes, by employing implantable defibrillators. However, HCM at earlier stages should be considered as a form of acquired combined channelopathy [137] rather than a structural disease, and should be treated as such, using provoking tests to identify patients at risk and employing targeted pharmacological antiarrhythmic therapies in selected patients. Moreover, considering that Ca^2+^ handling dysregulation seems to precede the effective presentation of the structural alterations of the disease, including hypertrophy and fibrosis, a preventative rather than corrective therapeutic approach could represent a major improvement in the therapeutic management of HCM [138]. Considering that SCD due to ventricular fibrillation represents the most unpredictable and devastating consequence of HCM, the fine comprehension of the different pathomechanisms underpinning arrhythmogenesis could help to improve and ameliorate the therapeutic management of HCM patients, through a better selection of high-risk patients to be subjected to aggressive arrhythmia-preventing strategies at early stages, either with drugs or with implantable devices. In this regard, hiPSC-CMs represent an innovative platform that could provide a significant advancement to HCM research; since hiPSC-CMs can be non-invasively obtained from different HCM patients, they could allow a deeper comprehension of mutation-specific arrhythmogenic pathomechanisms that may be the target of selective personalized preventive strategies.

## Figures and Tables

**Figure 1 cells-10-02789-f001:**
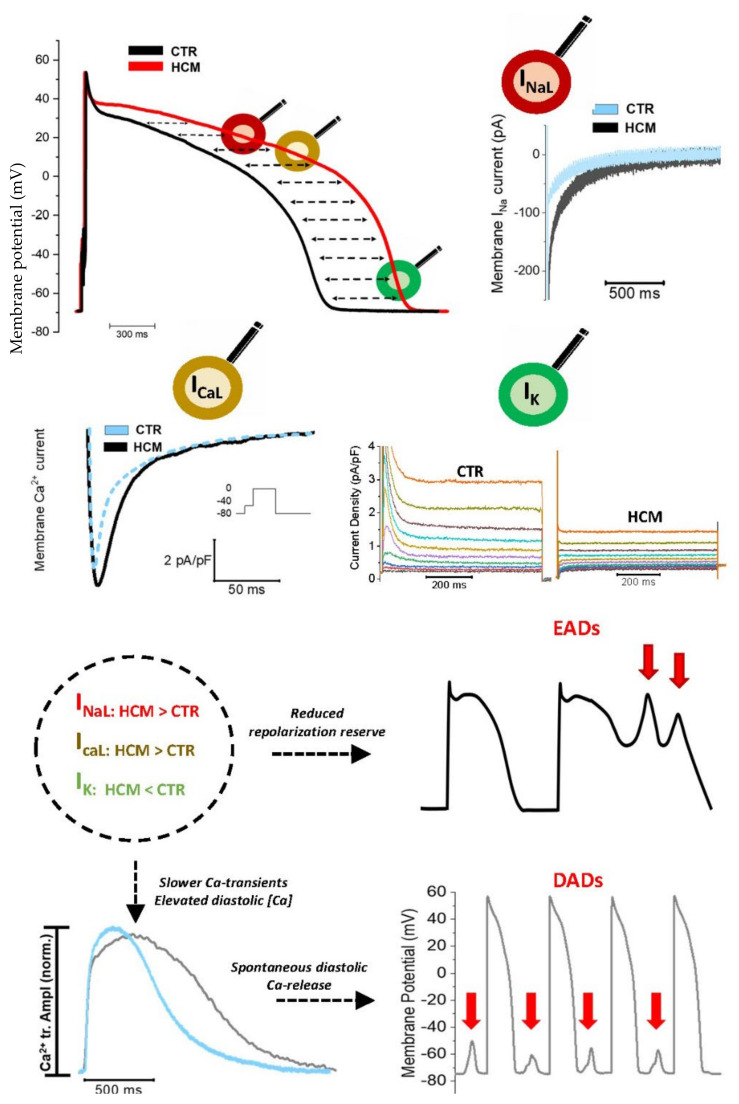
Ion channel remodeling in HCM cardiomyocytes. APD is consistently prolonged in HCM myocytes, due to a combination of increased I_NaL_, increased I_CaL_ amplitude, slower I_CaL_ inactivation and decreased potassium currents. APD prolongation raises the likelihood of EADs. Accumulation of intracellular calcium and Ca-overload is associated with enhanced spontaneous Ca-release from the SR, thus increasing the probability of calcium waves and DADs. Traces modified from Coppini et al. [26].

**Figure 2 cells-10-02789-f002:**
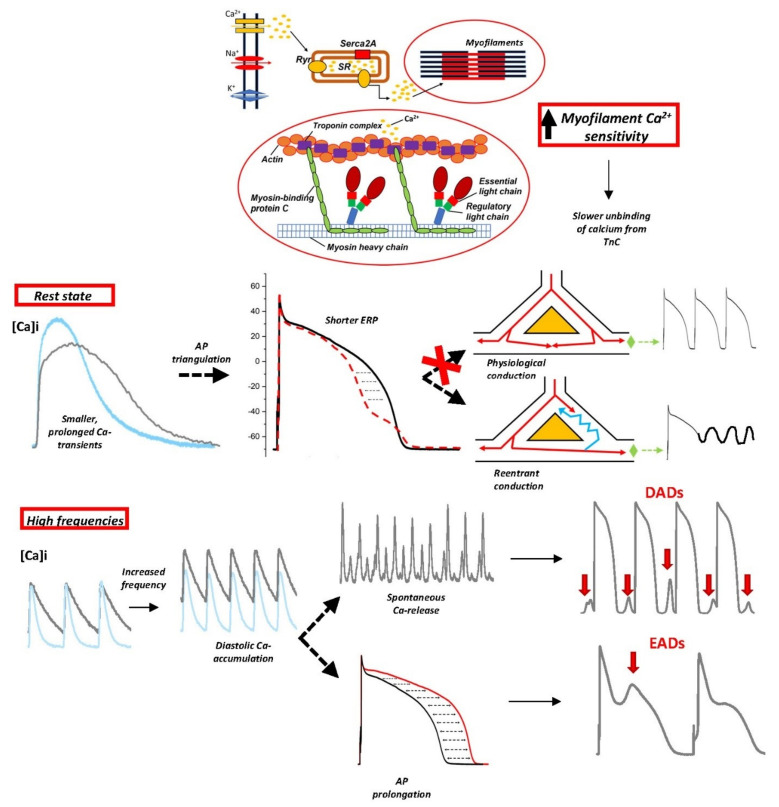
Myofilament Ca^2+^ sensitization as an arrhythmogenic mechanism in HCM. The increased sensitivity to Ca of the troponin complex in HCM cardiomyocytes causes a more prolonged binding of Ca to TnC and a slower detachment of the ion. This affects Ca-transients, which tend to be lower in amplitude and slower in their decay phase. This behavior is associated with triangulation of the AP, resulting in local shortening of the effective refractory period (ERP), rendering the tissue more susceptible to reentry arrhythmias. Moreover, the accumulation of diastolic Ca at higher pacing rates sensitizes the ryanodine receptors, thus increasing the likelihood of spontaneous diastolic openings, calcium waves and thus late-phase EADs and DADs. A sudden prolongation of APs due to Ca accumulation further raises the probability of afterdepolarizations. Traces modified from Coppini et al. [26], Coppini et al. [56], and Ferrantini et al. [40].

**Figure 3 cells-10-02789-f003:**
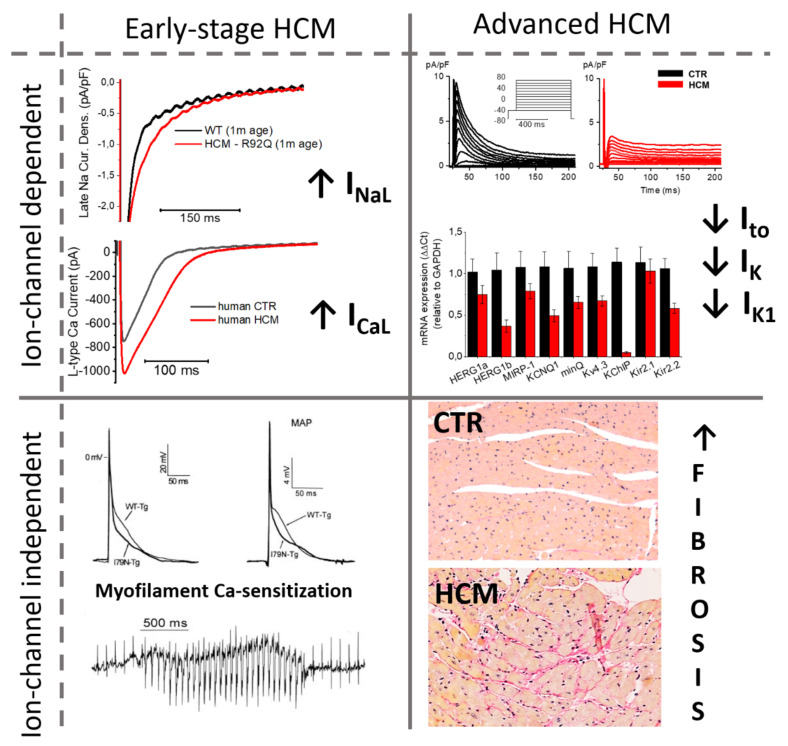
Channel-dependent and channel-independent mechanisms at different stages of disease. The principal mechanisms of arrhythmias in HCM are divided depending on the disease stage (early and advanced) and on the nature of such mechanisms (whether they depend or not on changes of ion currents). While the increase of I_NaL_ and I_CaL_ occur early in the pathogenesis of HCM, being the consequence of post-translational channel modifications (e.g., increased phosphorylation by CaMKII), the decrease of K currents occurs later, as it follows a profound change of cardiomyocyte gene expression profile, with reduced transcription of K current genes. Myofilament Ca sensitization may determine arrhythmias even in the absence of ion current changes, due to a combination of functional mechanisms that are detailed in the text. Finally, at later stages of the disease, replacement fibrosis, as well as local ischemia due to microvascular dysfunction, facilitate the formation of stable reentry circuits and underlie sustained arrhythmias. Traces modified from Knollman et al. [37], Coppini et al. [26] and, Coppini et al. [56].

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
