# Peer review of "Ion Channel Impairment and Myofilament Ca2+ Sensitization: Two Parallel Mechanisms Underlying Arrhythmogenesis in Hypertrophic Cardiomyopathy"

_cells, 2021, doi:10.3390/cells10102789_

Round 1
Reviewer 1 Report
In the present review, Santini and co-authors covered the tentative cellular mechanisms of HCM-associated arrhythmias. Briefly, they detailed some action potential abnormalities, describing the principal ion current alterations and the involvement of the myofilament calcium sensitization. However, there are weaknesses in the review that need to be addressed:
- Could the authors clarify, how this manuscript differs from the previous review by the same authors published a couple of months ago in Cardiovasc Res (https://doi.org/10.1093/cvr/cvaa124)
- Throughout the manuscript, there are multiple redundancies between the chapters that were initially divided into animal models and human cells. For instance, when descripting the identical HCM models in mice and human, line 220ff and line 366ff. It would highly improve the understanding of the manuscript, when the authors would summarize the findings for particular HCM models gathered from both animal and human studies in one chapter, rather than repeating nearly identical sentences.
- Since the authors focused on CAMKII as one of the modulators of HCM and arrhythmogenesis, it would be informative to provide knowledge about cause or consequence of APs and HCM. The authors could incorporate some evidence in the discussion.
- Throughout the manuscript, citations are partially missing, e.g. line 84-87, or cited references do not reflect the statements, e.g. line 107-109.
- Line 58-59: repetition from previous sentences.
- Line 64: “this highlighted…“, please stick to one tense.
- Line 92: tachy-arrhythmias is uncommon, rather: tachycardia / ventricular arrhythmia
- Line 103: which aforementioned pathways are meant here?
- Line 138: “Thanks to partnership with our local unit…”, it is rather untypical for a review to include acknowledgements in the main text.
- Line 183: what is meant by M cells?
- Line 220: “G>A in last nucleotide of exon 6”, please use the correct nomenclature for gene mutations.
- Line 224: “confirming that 1”, what is meant by 1 here?
- Line 250: mutation IN troponin-T
- Line 510-518: mosty repetition from previous chapter. Please harmonize.
- Line 586: wet biology is uncommon, rather: wet lab
- Figure 1: legend at y-axis in first graph is missing. Further, figure is quite similar to previous publication from authors (https://doi.org/10.1093/cvr/cvaa124)
- Throughout the entire manuscript: please introduce abbreviations before first use (e.g. line 22 EAD, line 96 AP, line 215 CTRL, etc.)
- Throughout the entire manuscript, please check and homogenize spacing (e.g. line 40, 54, 224, etc), spacing between units (e.g. line 123), punctuation (line 57, 87, etc.), spelling (e.g. line 100 wherean, line 107 is comprises, line 459 kinetics kinetics, etc.), consistence (patho-mechanism vs pathomechansim)
Author Response
We thank the reviewer for his/her kind suggestions. We are providing a point-by-point response in the attached file

Reviewer 2 Report
This is a well written manuscript that summarizes the relationship between myofilament Ca2+ sensitivity and the development of arrhythmias in HCM hearts. The authors compiled information about genes related to HCM as well as different biological models studied to date, i.e., explanted human tissue, mouse models and hiPSC-CM. I have a few recommendations for the authors to address:
- They should include TNNC1 in the list of HCM genes (J Mol Cell Cardiol. 2020 May;142:118-125. doi: 10.1016/j.yjmcc.2020.04.005. Epub 2020 Apr 9. PMID: 32278834). In addition, pathogenic variants in TNNC1 have been associated with arrhythmogenesis (J Biol Chem. 2012 Sep 14;287(38):31845-55. doi: 10.1074/jbc.M112.377713. Epub 2012 Jul 18. PMID: 22815480). Landstrom and Ackerman groups have been instrumental in this determination.
- Subtopic 2.3 – hiPSC: they should include and discuss publications from Knollmann’s group with hiPSC-CM I79N (J Am Coll Cardiol. 2017 Nov 14;70(20):2600-2602. doi: 10.1016/j.jacc.2017.09.033. PMID: 29145956 and J Mol Cell Cardiol. 2018 Jan;114:320-327. doi: 10.1016/j.yjmcc.2017.12.002. Epub 2017 Dec 5. PMID: 29217433).
- The discussion surrounding PKA would benefit from the following article that showed long term lack of TnI phosphorylation by PKA induces EC uncoupling (J Biol Chem. 2014 Aug 15;289(33):23097-23111. doi: 10.1074/jbc.M114.561472. Epub 2014 Jun 27. PMID: 24973218)
Author Response

(The authors gave the same response as above.)

Round 2
Reviewer 1 Report
- The authors argue that CAMKII may be a cause of HCM due the Ca2+ sensitization. However, CAMKII activation it is not exclusively activated by increased cytosolic calcium. Moreover, CAMKII can be activated by elevated ROS production in a calcium-independent way (https://doi.org/10.1016/j.cell.2008.02.048). Considering that ROS production also plays a key role in HCM and arrhythmogenesis (doi: 10.1089/ars.2017.7236), and furthermore, facilitates the CAMKII Ca2+ sensitization (https://doi.org/10.1161/CIRCRESAHA.110.221911), could the authors elaborate about this point?
- Some other reports showed that CAMKII plays a key role in myofilament Ca2+ desensitization (doi: 10.1016/j.ceca.2015.08.001). Could CAMKII plays a double role promoting an ion channel activation in one site, and myofilament Ca2+ desensitization in the another? May these changes be expected in an early or onset-stage of arrhythmogenesis in hypertrophic cardiomyopathy?
- Although slightly modified, there is still a large amount of redundant sentences within the manuscript. Further, the authors summarized the advantage and disadvantage of the different models for studying arrhythmogenesis in hypertrophic cardiomyopathy in great detail in their previous publication.
- Nomenclature for gene mutations is not homogenized for alpha-Tropomyosin.
Author Response
We thank the reviewer for the comments. Please see the attached file

Reviewer 2 Report
Thank you for carefully addressing my comments. I look forward to seeing the manuscript published.
Author Response
We thank the reviewer for his/her comments.